# Perioperative Geriatric Assessment as A Predictor of Long-Term Hepatectomy Outcomes in Elderly Patients with Hepatocellular Carcinoma

**DOI:** 10.3390/cancers13040842

**Published:** 2021-02-17

**Authors:** Masaki Kaibori, Hideyuki Matsushima, Morihiko Ishizaki, Hisashi Kosaka, Kosuke Matsui, Asao Ogawa, Kengo Yoshii, Mitsugu Sekimoto

**Affiliations:** 1Department of Surgery, Hirakata Hospital, Kansai Medical University, Hirakata, Osaka 573-1191, Japan; h.matsushima0921@gmail.com (H.M.); ishizakm@hirakata.kmu.ac.jp (M.I.); kosakahi@hirakata.kmu.ac.jp (H.K.); matsuik@hirakata.kmu.ac.jp (K.M.); sekimotm@hirakata.kmu.ac.jp (M.S.); 2Next-Generation Minimally Invasive Surgery, Kansai Medical University, Hirakata, Osaka 573-1191, Japan; 3Department of Psycho-Oncology, National Cancer Center East Hospital, Kashiwa 277-8577, Japan; asogawa@east.ncc.go.jp; 4Department of Mathematics and Statistics in Medical Sciences, Kyoto Prefectural University of Medicine, Kyoto 606-0823, Japan; yoshii-k@koto.kpu-m.ac.jp

**Keywords:** geriatric assessment, elderly patients, recurrence-free survival, overall survival, hepatectomy, liver cancer

## Abstract

**Simple Summary:**

Older patients are considered to have increased risk for complications and survival after major surgery, but age alone is not a reliable predictor of post-operative complications and outcomes. To date, no universal screening test adequately predicts postoperative outcomes in older patients. This retrospective study recorded pertinent baseline geriatric assessment variables to identify risk factors for recurrence-free survival (RFS) and overall survival (OS) in hepatocellular carcinoma (HCC) for patients aged ≥70 years who undergo hepatectomy. The change of geriatric 8 (G8) at six months postoperatively was the most significant predictive factor for RFS and OS among various geriatric assessments. G8 score is a useful screening method for older HCC patients who qualify for elective liver resection.

**Abstract:**

This retrospective study recorded pertinent baseline geriatric assessment variables to identify risk factors for recurrence-free survival (RFS) and overall survival (OS) after hepatectomy in 100 consecutive patients aged ≥70 years with hepatocellular carcinoma. Patients had geriatric assessments of cognition, nutritional and functional statuses, and comorbidity burden, both preoperatively and at six months postoperatively. The rate of change in each score between preoperative and postoperative assessments was calculated by subtracting the preoperative score from the score at six months postoperatively, then dividing by the score at six months postoperatively. Patients with score change ≥0 comprised the maintenance group, while patients with score change <0 comprised the reduction group. The change in Geriatric 8 (G8) score at six months postoperatively was the most significant predictive factor for RFS and OS among the tested geriatric assessments. Five-year RFS rates were 43.4% vs. 6.7% (maintenance vs. reduction group; HR, 0.19; 95%CI, 0.11–0.31; *p* < 0.001). Five-year OS rates were 73.8% vs. 17.8% (HR, 0.12; 95%CI, 0.06–0.25; *p* < 0.001). Multivariate Cox proportional hazards analysis showed that perioperative maintenance of G8 score was an independent prognostic indicator for both RFS and OS. Perioperative changes in G8 scores can help forecast postoperative long-term outcomes in these patients.

## 1. Introduction

Many countries are experiencing an increase in life expectancy, which has led to the worldwide clinical concern in management of malignant diseases in the elderly [1,2,3]. There is a high incidence of comorbid illnesses in elderly patients and they are considered to have high-risk for major surgery [3,4]. However, postoperative complications and outcomes cannot be predicted by age alone [5]. Functional status and comorbidity can be measured by other physiologic parameters to identify patients who are likely to experience postoperative complications or post-discharge institutionalization [6]. Furthermore, the Comprehensive Geriatric Assessment (CGA) is a multidimensional tool that can guide health management for older patients; it can identify patients with higher risk of adverse outcomes through evaluating comorbidities, nutritional status, cognitive status, functional status, and geriatric syndromes [7]. 

Hepatocellular carcinoma (HCC) is common and expected to become even more common in elderly patients over time. We recently performed a large nationwide study [8], which indicated that elderly patients aged ≥ 75 years had significantly worse overall survival (OS) after hepatectomy for HCC compared with middle-aged and young patients; however, the differences in recurrence-free survival (RFS) among the three groups were relatively small. The cumulative incidence of other causes of death among the elderly patients was significantly different from the incidences of HCC-related or liver-related death among the three groups. Age itself should not be a contraindication for hepatic resection treatment of HCC, but the findings suggested that elderly patients with comorbidities should be more stringently selected for surgery. Therefore, preoperative liver function tests alone are not sufficient to determine the indications for surgical resection in elderly HCC patients; a comprehensive evaluation of physical and mental functions is essential.

The CGA can identify reversible conditions that can be addressed to improve patient fitness prior to surgery. This evaluation might also help clinicians to anticipate complications and long-term survival by predicting the need for additional support during and after surgery. In addition, we previously reported that the Geriatric 8 (G8) score, partially based on nutritional assessment, is a useful screening parameter for older HCC patients who qualify for elective liver resection. Preoperative G8 scores can effectively predict postoperative complications in older HCC patients [9]. Here, we performed a prospective study involving baseline geriatric assessment variables to identify prognostic factors for postoperative long-term outcomes in HCC patients aged ≥70 years who were undergoing hepatectomies.

## 2. Results

### 2.1. Patients’ Perioperative Characteristics 

There were 100 patients in this study (median age, 77 years; range, 70–92 years). Their baseline demographic and clinical characteristics, CGA results, and surgical procedures are summarized in Table 1. Overall preoperative CGA results were Geriatric Depression Scale score ≥3 in 50% of the patients, Charlson Comorbidity Index ≥ 4 in 51%, MMSE < 28 in 47%, Barthel index of 100 in 82%, vitality index of 10 in 75%, IADL score ≥ 5 in 70%, VES13 score ≥ 2 in 50%, and G8 score < 13 in 34%. Ninety-eight patients (98%) had comorbidities, including 48 patients with hypertension, 31 patients with diabetes mellitus, and 30 patients with a history of other cancers. The history of cancer in these 30 patients was as follows: gastric cancer (5 cases), colon cancer (5 cases), prostate cancer (5 cases), bladder cancer (3 cases), lung cancer (2 cases), pancreatic cancer (2 cases), laryngeal cancer (2 cases), and other cancers (6 cases). In terms of surgical procedures, anatomic resection was performed in 55% of the patients, while the laparoscopic approach was implemented in 22% of the patients. Postoperative complications developed in 19 patients (19%): 10 had refractory pleural effusion and/or ascites, 2 had intra-abdominal abscesses, 3 had deep incisional surgical site infection, 2 had ileus, and 2 had other complications. Clavien–Dindo grades I, II, IIIa, and IIIb complications occurred during postoperative hospital stays in 5%, 6%, 7%, and 1% of patients, respectively.

### 2.2. Changes in Perioperative Geriatric Assessments

Figure 1 shows perioperative changes in eight geriatric assessments between the maintenance and reduction groups. There was a significant difference between the two groups in all assessments, except for the MMSE. The largest difference among all assessments was found in the G8 score between the two groups. In the reduction group, the mean G8 score decreased from 13.8 preoperatively to 11.4 at six months postoperatively, compared to the maintenance group, in which the mean G8 score increased from 13.0 to 14.5 during that same time period; the difference between groups at the six month-postoperative time point was statistically significant (*p* < 0.001; Figure 1H).

Table 2 shows the results of a multivariate statistical model incorporating perioperative changes in eight geriatric assessments. The rate of change in each score between preoperative and postoperative assessments was calculated by subtracting the preoperative score from the score at six months postoperatively, then dividing by the score at six months postoperatively. Patients with score change ≥ 0 comprised the maintenance group, while patients with score change < 0 comprised the reduction group. Multivariate Cox proportional hazards analysis identified three independent prognostic predictors for RFS: reduced Charlson Comorbidity Index score (hazard ratio, 3.53; 95% CI: 2.03–6.11; *p* < 0.001), reduced vitality index score (hazard ratio, 0.40; 95% CI: 0.19–0.82; *p* = 0.012), and reduced G8 score (hazard ratio, 0.19; 95% CI: 0.11–0.31; *p* < 0.001). Cox analysis also identified three independent prognostic predictors for OS: reduced geriatric depression scale score (hazard ratio, 2.49; 95% CI: 1.19–5.21; *p* = 0.016), reduced vitality index score (hazard ratio, 0.24; 95% CI: 0.10–0.57; *p* = 0.001), and reduced G8 score (hazard ratio, 0.12; 95% CI: 0.06–0.25; *p* < 0.001).

Figure 2 shows a comparison of long-term outcomes in the G8 score between the two groups. Recurrence of HCC occurred in eight patients in the reduction group and in two patients in the maintenance group by six months post-surgery. Each patient underwent transarterial chemoembolization, radiofrequency ablation, or repeat hepatic resection for recurrence of HCC (4 of 10 patients received treatment at six months after surgery, and the remaining six patients were treated after six months). The median follow-up periods were 50.5 and 24.1 months in the maintenance and reduction groups, respectively. The five-year RFS rates were 43.4% and 6.7% in the maintenance and reduction groups, respectively (hazard ratio, 5.35; 95% CI, 3.18–9.01; *P* < 0.001; Figure 2A). The five-year OS rates were 73.8% and 17.8% in the maintenance and reduction groups, respectively (hazard ratio, 8.09; 95% CI: 4.03–16.27; *P* < 0.001; Figure 2B).

### 2.3. Perioperative Characteristics of HCC Patients Classified According to G8 Change 

Table 3 summarizes the perioperative characteristics of both groups. Platelet count, prothrombin activity, serum albumin, total bilirubin, alpha-fetoprotein (AFP), and PIVKA-2 levels were compared between the two groups before and six months after surgery. No differences were detected between groups in terms of age, sex, diabetes mellitus status, presence or absence of esophageal and/or gastric varices, hepatitis B surface antigen status, hepatitis C virus antibody, Child–Pugh class, indocyanine green retention rate at 15 min, peripheral blood count, general biochemical blood laboratory test results, conventional liver function test results, or serum PIVKA-II concentrations. The serum AFP concentration significantly differed between groups both before surgery and six months after surgery. No differences were detected between groups in terms of operative characteristics, pathological features, or postoperative characteristics (Table 3). The maximum tumor size was significantly greater in the reduction group than in the maintenance group.

Table 4 shows changes in each of the eight G8 items between groups. Loss of appetite, weight loss, mobility, body mass index (BMI), and feelings of personal health all significantly declined in the reduction group compared to the maintenance group.

### 2.4. Univariate and Multivariate Analysis of Prognostic Factors for Long-Term Survival

Cox proportional hazards analysis revealed three independent prognostic predictors for RFS: tumor size ≥3.5 cm (hazard ratio, 2.98; 95% CI, 1.39–6.41; *P* = 0.005), positive surgical margin invasion (hazard ratio, 4.59; 95% CI, 1.27–16.68; *P* = 0.02), and G8 score maintenance (hazard ratio, 0.16; 95% CI, 0.08–0.34; *P* < 0.001). Cox analysis also revealed three independent prognostic predictors for OS: serum albumin ≥4.0 g/dL (hazard ratio, 0.29; 95% CI, 0.10–0.85; *P* = 0.025), tumor size ≥ 3.5 cm (hazard ratio, 3.59; 95% CI, 1.12–11.58; *P* = 0.032), and G8 score maintenance (hazard ratio, 0.08; 95% CI, 0.03–0.22; *P* < 0.001; Table 5).

## 3. Discussion

In a large nationwide study in Japan, we previously found that elderly patients aged ≥ 75 years had significantly better RFS and OS after hepatic resection for small HCC than after radiofrequency ablation, microwave ablation, or transcatheter arterial chemoembolization [8]. Thus, hepatic resection may be considered a first-line treatment modality for a single small HCC tumor; this is consistent with Japanese clinical practice guidelines [10]. Overall, surgery has tended to yield superior RFS and OS rates, compared with other treatments, for elderly HCC patients.

Japan is considered an “aging society” and there is a high incidence of death due to cancer in elderly people [11]. In the next 20 years, the number patients with cancer who are elderly is projected to increase by 67% [12]. Elderly patients with cancer are likely to find it difficult to self-manage, control psychological symptoms, and manage complex medical information [11]. Thus, these vulnerable patients should be evaluated and careful consideration should be taken in selecting treatment strategies for them [13]. 

To investigate the effects of markers of frailty on the postoperative course of disease, G8 tools have recently been applied to patients undergoing general surgery and surgical oncology treatments [14,15,16]. Chronological age alone has never been a reliable predictor of postoperative complications or survival after cancer surgery; methods to assess physiologic age might be more helpful in predicting patient outcomes. In surgical and non-surgical studies, the G8 tool and Flemish version of the Triage Risk Screening Tool have been shown to provide valuable information concerning elderly patients with cancer by predicting functional decline and OS [14]. The G8 tool is convenient, easy, and can be administered rapidly. It is generally completed within 5 minutes and can often be administered by a nurse without specific expertise in geriatrics. The G8 tool comprises a set of questions that evaluate functional status, mental status, psychological state, comorbidities, medical history, and nutritional status in an older individual (typically aged ≥ 65 years). We investigated the G8 tool as a predictor for RFS and OS in elderly patients with HCC who underwent hepatic resection. Perioperative patient assessment with the G8 tool may identify factors associated with adverse postoperative events and increased resource utilization; this information would enable the facilitation of strategies to address and minimize such risks. Better estimating postoperative recurrence and survival could yield improvements in understanding the risk-benefit ratio for the patient and inform surgical decision-making for the healthcare provider. This could lead to modified treatment sequences, including decisions about proceeding with chemotherapy or radiation before surgery. Improvements through focused effort in postoperative care for elderly patients are reportedly beneficial after cancer surgery; application of the CGA as a perioperative assessment component could facilitate interventions during both pre- and postoperative periods [17]. In this study, the change in G8 score at six months postoperatively was the most significant prognostic factor for RFS and OS among the tested geriatric assessments (Table 2). The group with the reduced postoperative G8 score had a greater median serum AFP concentration and a larger tumor size (Table 3). Many researchers, including us, have identified preoperative serum AFP concentration, tumor size, and Child–Pugh score as prognostic factors for recurrence and survival after surgery for HCC [18,19,20]. Therefore, we consider it necessary to re-evaluate the results of this study using statistical methods such as propensity score matching or inverse probability of treatment weighting in order to eliminate bias between the two groups. However, this type of examination was not possible due to the small number of patients in each group in our study. Therefore, in the future, we plan to examine a larger cohort as a multicenter collaborative-research study. 

In this study, we suspect that perioperative G8 score reduction might constitute a surrogate marker for prediction of long-term survival. In this study, the loss of appetite, weight loss, mobility, BMI, and feelings of my health all significantly declined in the reduction group, compared with the maintenance group (Table 4). Three of the eight items in the G8 tool involve nutritional status. Perioperative G8 score reduction in HCC patients may indicate the importance of nutritional therapy focused on food intake volume, weight loss, and BMI. Seven of the eight items in the G8 tool may be affected by perioperative therapeutic interventions. 

We previously found that patients with HCC and hepatic impairment experienced reductions in body mass, fat mass, and insulin resistance at six months after liver resection in response to increased exercise. Postoperative physical strength can be maintained by perioperative exercises and lead to patients resuming their daily activities sooner [21,22]. We recently began a clinical trial in elderly patients to investigate exercise for six months postoperatively. The aim of the study is to improve neuropsychological status by increasing food intake and maintaining body weight with enhanced muscle mass and reduced fat mass.

Postoperative G8 score reduction was identified as an independent predictor of RFS and OS in multivariate analysis (Table 5). This implied that older HCC patients with postoperative G8 score maintenance had a good possibility of uneventful recovery after surgery, while postoperative recurrence and death could be reliably predicted for the group with G8 score reduction. No current universal screening test adequately predicts postoperative long-term survival in at-risk older patients. In this study, HCC recurred in eight patients in the reduction group and two patients in the maintenance group by six months after surgery. Each patient underwent transarterial chemoembolization, radiofrequency ablation, or repeat hepatic resection for recurrence of HCC (4 of 10 patients received treatment at six months after surgery). The OS is affected by the ability to perform curative treatment for recurrent HCC. Treatments for recurrent HCC may worsen the patient’s general condition, including their nutritional status. It is therefore important to note that our data include deterioration of postoperative G8 values due to recurrence of HCC and its treatment. Further studies of postoperative survival in patients with HCC who undergo liver resection are needed to confirm our findings. 

## 4. Materials and Methods

### 4.1. Patients

Between January 2014 and September 2018, we screened all patients scheduled for liver resections at the Hospital of Kansai Medical University (Osaka, Japan). Inclusion criteria included the following: planned elective hepatectomy, 18–95 years of age, presence of adequate renal and cardiopulmonary function, and written informed consent. A single surgeon who had performed more than 1,500 hepatic resections was responsible for all procedures in this study. A total of 230 patients with HCC underwent R0 resections, which were defined as the macroscopic removal of all tumors. Seventy-nine patients were aged <70 years on admission, 11 patients died within six months postoperatively (three patients ≥70 years, eight patients <70 years), 29 patients aged ≥70 years declined to be enrolled in this study or were unable to complete a six-month postoperative survey, and 11 patients were followed up for fewer than six months postoperatively. The remaining 100 patients aged ≥70 years, all of whom were followed up for more than six months after hepatectomy, were included in the retrospective review of prospectively collected data (Appendix A). None of the patients received preoperative or postoperative adjuvant therapy. All patients provided written informed consent to participate in this study. The study protocol was approved by the institutional ethics committee of Kansai Medical University (reference number: KMU 2014902).

### 4.2. Clinicopathologic Variables, Treatment Algorithm for HCC, and Surgical Procedures 

Patients were measured for the indocyanine green retention rate at 15 min (ICGR15) and underwent conventional liver function tests prior to surgery. Hepatitis B surface antigen and hepatitis C antibody were measured for hepatitis screening. Patients also received measures for levels of AFP and protein induced by vitamin K absence/antagonism-II (PIVKA-II). 

We used the updated treatment algorithm for HCC, which included a combination of five factors: liver function reserve, extrahepatic metastasis, vascular invasion, tumor number, and tumor size [10]. We evaluated liver functional reserve based on the Child–Pugh classification. When hepatectomy was being considered, the degree of liver damage (including a measurement of ICGR15) was used to make a decision. We have summarized the new treatment algorithm as follows. HCC patients with Child–Pugh A/B liver function without extrahepatic metastasis or vascular invasion are recommended to receive one of three treatment regimens. First, either surgical resection radiofrequency ablation is recommended with no priority for up to three HCCs measuring ≤3 cm, or surgical resection is recommended as first-line therapy for solitary HCC regardless of size. Second, for up to three HCCs measuring >3 cm, surgical resection is recommend as first-line therapy, and transarterial chemoembolization is recommended as second-line therapy. Third, for patients with HCC accompanied by vascular invasion without extrahepatic metastasis, a combination of embolization, hepatectomy, hepatic arterial infusion chemotherapy, and molecular targeted therapy is recommended. Treatment is selected for each patient according to the individual situation, including consideration of the following factors: liver function, the condition of HCC, and the extent of vascular invasion. 

The Brisbane terminology proposed by Strasberg et al. was used to classify surgical procedures [23]. Anatomic resection was defined as resection of the tumor together with the related portal vein branches and corresponding hepatic territory. Anatomic resection was classified as hemihepatectomy (resection of half of the liver), extended hemihepatectomy (hemihepatectomy plus removal of additional contiguous segments), sectionectomy (resection of two Couinaud subsegments [24]), or segmentectomy (resection of one Couinaud subsegment). All other non-anatomic procedures were classified as limited resections. Limited resection was used to manage both peripheral and central tumors. Because partial hepatectomy allows adequate surgical margins, it was used to manage peripheral tumors and those with extrahepatic growth. Conversely, because of the difficulty and risks associated with achieving adequate margins, enucleation was used to manage central tumors near the hepatic hilum or major vessels. Each specimen was reviewed by a senior pathologist who performed a histological review to confirm the final diagnosis.

### 4.3. Perioperative Geriatric Assessment Measurements 

Patients underwent CGA measurements both preoperatively and at six months postoperatively. These CGA studies were conducted by two medical assistants who were familiar with medical information. Any contents of the GA that could not be confirmed at the time of data collection by the assistant were validated by the attending physician, who determined the final score. Assessments of baseline cognition, nutritional, and functional statuses, and comorbidity burdens were performed. 

The Barthel Index, which comprises 10 questions pertaining to ADL (e.g., feeding, mobility, and grooming), was used to evaluate basic activity of daily living (BADL) [25]. All scores were weighted, and the maximum score was 100 points (100 points = PS0). The Japanese author-certified version of the Mini-Mental State Examination (MMSE-J; purchased from the publisher (Nihon Bunka Kagakusha, Tokyo, Japan)) was used to assess cognitive status at hospital admission [26]. The maximum test score is 30; the lower cut-off limit is 23/24. A lower score is considered indicative of functional cognitive decline. The geriatric depression scale (GDS)-15, which assesses depression in elderly patients [27], was used to assess emotions and moods; the questions were identical to those included in the GDS-Short Form (SF) [28]. This yes/no questionnaire comprises 15 questions concerning feelings and moods. A score ≥5 is considered indicative of a tendency toward depression, while a score ≥10 is considered indicative of depressive symptoms. 

The validated Japanese version of the IADL scale (Lawton and Brody) [29] has demonstrated positive correlations with MMSE and GDS assessments [30]. This scale evaluates various characteristics of ADL, including (1) using the telephone, (2) shopping, (3) preparing meals, (4) housework, (5) laundry, (6) using transportation, (7) managing medication, and (8) managing property. The maximum scores are five for men (excluding abilities 2, 3, and 4) and eight for women and thus points were converted into a percentage of the total to adjust for sex-based differences. The vitality index was originally developed as a measure of activity among elderly people with disabilities in Japan [31]. This index evaluates five activities: (1) getting up, (2) communicating, (3) feeding, (4) toileting, and (5) rehabilitation or other activities. The maximum score is 10, and the scores are weighted. A score reduction of 1 point is considered indicative of reduced volition.

To screen community-dwelling populations and identify older persons at risk of impending health deterioration, we used the Vulnerable Elderly Survey (VES-13), which is a function-based tool. Through a summed score, the VES-13 includes considerations of age, self-rated health, physical function limitations, and functional disabilities. Poor outcome risk is indicated by scores of ≥3 [32]. Comorbidity burden was quantified using the Charlson Comorbidity Index, which contains 19 comorbidity categories and assigns a weighted value to each comorbidity based on the corresponding risk of 1-year mortality [33]. The G8 is a screening tool that includes seven items from the MNA and an age-related item (<80, 80–85, or >85 years; Table 5), with final scores of 0–17. A score <14 is considered indicative of a geriatric risk profile [14,34]. All of these preoperative items were classified into two groups (maintenance and reduction) using the median value of our 100 patients.

The rate of change in each score between preoperative and postoperative assessments was calculated by subtracting the preoperative score from the score at six months postoperatively, then dividing by the score at six months postoperatively. Patients with score change ≥0 comprised the maintenance group, while patients with score change <0 comprised the reduction group. 

We classified the perioperative characteristics, including the CGA components of HCC patients of the two groups, according to the G8 change.

### 4.4. Follow-Up

To determine morbidity and mortality following hepatectomy, we recorded peri- and postoperative complications and deaths. All surviving patients were followed up at three-month intervals after discharge. Follow-up assessments included physical examination; liver function tests; chest radiography to identify any pulmonary metastases; and ultrasonography, computed tomography, or magnetic resonance imaging to identify any intrahepatic recurrence. If chest radiography assessment revealed any abnormalities, chest computed tomography was performed; bone scintigraphy was used to diagnose bone metastases.

In cases where HCC recurrence was detected on the basis of changes in tumor markers or on imaging assessments, recurrence limited to the remnant liver was treated by transarterial chemoembolization, lipiodolization, re-resection, or percutaneous local ablative therapy (e.g., radiofrequency ablation). For cases of extrahepatic metastases, active treatment and/or molecular targeting (e.g., via sorafenib) was prescribed in patients with good hepatic functional reserve (Child–Pugh class A or B) and good performance status (0 or 1); other patients received radiation therapy alone to relieve symptoms of bone metastases. Patients with a solitary extrahepatic metastasis and no intrahepatic recurrence received surgical resection.

### 4.5. Statistical Analysis

Clinical characteristics were compared between groups using either Wilcoxon’s rank-sum test, the chi-squared test, or Fisher’s exact test. Probabilities for RFS and OS according to changes in each geriatric assessment were calculated using the Kaplan–Meier method. Hazard ratios for RFS and OS and their 95% confidence intervals (CIs) were estimated using univariate Cox analysis. Multivariate analysis was performed using Cox proportional hazards analysis. The following variables were examined as potential prognostic predictors: sex, age, serum total bilirubin, albumin, prothrombin activity, platelet count, AFP, and PIVKA-II concentrations, tumor number, maximum tumor size, surgical margin invasion, and score changes in measurements such as the vitality index and G8. Differences between groups concerning perioperative changes in eight geriatric assessments were assessed using the chi-squared test or Fisher’s exact test. A two-sided *p*-value < 0.05 was considered statistically significant. All statistical analyses were performed with R version 4.0.3 (R Foundation for Statistical Computing, Vienna, Austria) using the survival and matching packages.

## 5. Conclusions

Our findings demonstrate that perioperative changes in G8 score, which is partially based on nutritional assessment, constitute a useful screening parameter for older HCC patients who qualify for elective liver resection. Perioperative changes in G8 score can help forecast postoperative long-term outcomes in these patients.

## Figures and Tables

**Figure 1 cancers-13-00842-f001:**
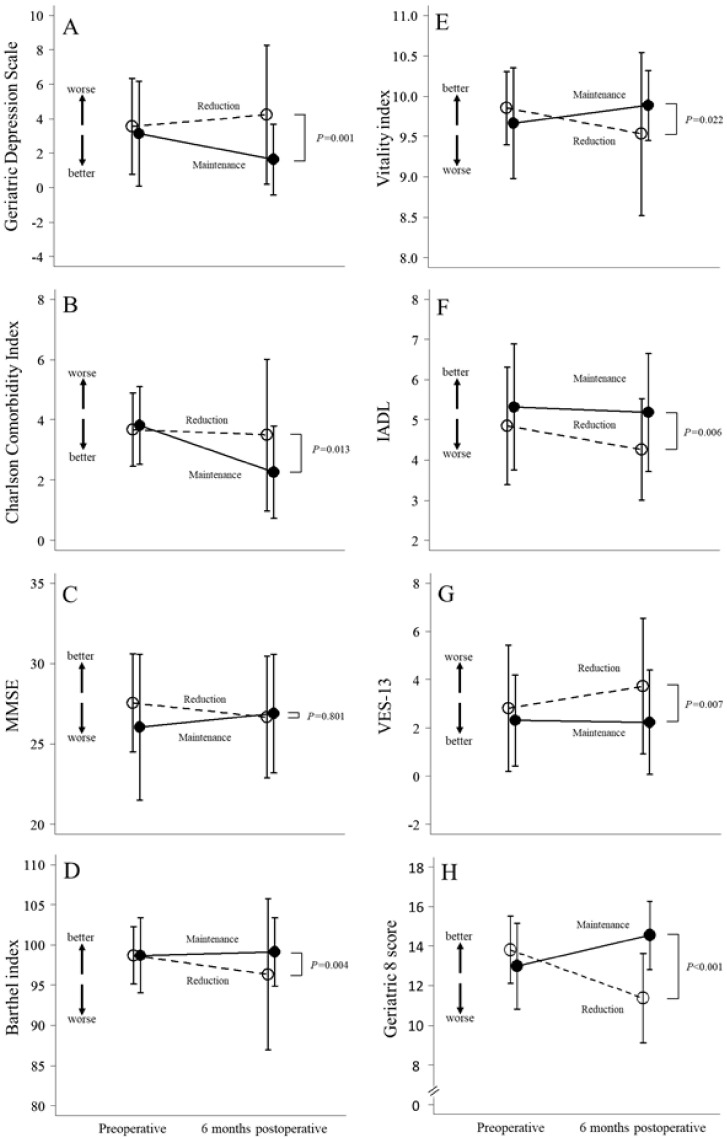
Perioperative changes in eight geriatric assessments between the maintenance and reduction groups.

**Figure 2 cancers-13-00842-f002:**
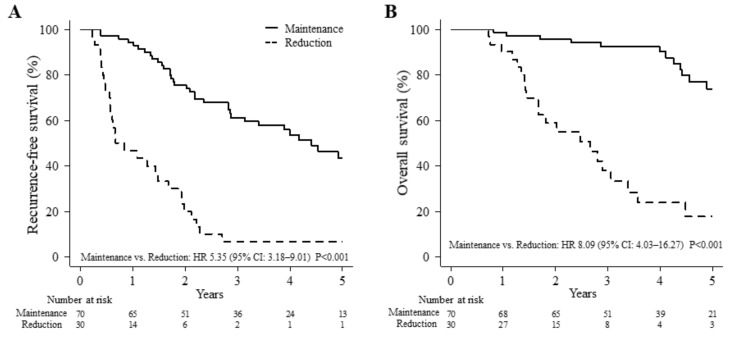
A comparison of long-term outcomes in the G8 score between the two groups.

**Table 1 cancers-13-00842-t001:** Perioperative characteristics of 100 patients aged ≥ 70 years who underwent hepatic resection for treatment of hepatocellular carcinoma.

Patients (*n* = 100)	*n* = 100 (%) or Median (5th Percentile, 95th Percentile)
Age (years)	77 (71, 87)
Age distribution	
70–75 years	34
76–80 years	42
81–85 years	17
86–90 years	5
≥91 years	2
Sex (male/female)	76/24
Body mass index (kg/m2)	23.4 (18.5, 28.9)
Preoperative liver function	
HBV/HCV/Alcoholic	14/41/19
Esophageal and/or gastric varices	8 (8%)
Child–Pugh class (A/B)	96/4
ICGR15 ≥ 18%	47 (47%)
Albumin level < 4.0 g/dL	47 (47%)
Total bilirubin level ≥ 0.8 mg/dL	52 (52%)
Platelet count < 16 × 104/mL	52 (52%)
AST level ≥ 35 IU/L	50 (50%)
Preoperative geriatric assessments	
Geriatric Depression Scale score ≥3	50 (50%)
Charlson Comorbidity Index ≥4	51 (51%)
MMSE score < 28	47 (47%)
Barthel index = 100	82 (82%)
Vitality index = 10	75 (75%)
IADL score ≥ 5	70 (70%)
VES13 score ≥ 2	50 (50%)
G8 score < 13	34 (34%)
Comorbidities	98 (98%)
Diabetes mellitus	31
Hypertension	48
Cerebrovascular disease	13
Myocardial disease or arrhythmia	21
Respiratory disease	8
Renal disease	6
Other cancers	30
Others	17
Surgical procedure	
Non-anatomic resection	45 (45%)
Anatomic resection	55 (55%)
Laparoscopic approach	22 (22%)
Extent of hepatic resection ≥ 2 sections	27 (27%)
Postoperative mortality	0 (0%)
Postoperative complications	19 (19%)
Refractory pleural effusion and/or ascites	10
Intra-abdominal abscess	2
Deep incisional SSI	3
Ileus	2
Others	2
Clavien–Dindo classification	
Grade I	5
Grade II	6
Grade IIIa	7
Grade IIIb	1
Grade IVa	0
Grade IVb	0

Data are shown as n (%) or median (5th percentile, 95th percentile). HBV: hepatitis B virus; HCV: hepatitis C virus; ICGR15: indocyanine green retention rate at 15 minutes; AST: aspartate aminotransferase; MMSE: Mini Mental State Examination; IADL: Instrumental Activity of Daily Living; VES: Vulnerable Elders Survey; SSI: surgical site infection.

**Table 2 cancers-13-00842-t002:** Hazard ratios of eight geriatric assessment components for recurrence-free survival and overall survival in HCC patients aged ≥ 70 years who underwent hepatic resection.

Variables		RFS			OS		
		HR	(95% CI)	*P*	HR	(95% CI)	*P*
Geriatric Depression Scale	reduction (vs. maintenance)	1.17	(0.70–1.95)	0.546	2.49	(1.19–5.21)	0.016
Charlson Comorbidity Index	reduction (vs. maintenance)	3.53	(2.03–6.11)	<0.001	1.88	(0.91–3.88)	0.090
MMSE	reduction (vs. maintenance)	0.87	(0.52–1.45)	0.588	0.85	(0.43–1.69)	0.648
Barthel index	reduction (vs. maintenance)	0.59	(0.28–1.26)	0.174	0.43	(0.17–1.05)	0.062
Vitality index	reduction (vs. maintenance)	0.40	(0.19–0.82)	0.012	0.24	(0.10–0.57)	0.001
IADL	reduction (vs. maintenance)	0.97	(0.54–1.73)	0.917	0.66	(0.31–1.38)	0.268
VES-13	reduction (vs. maintenance)	0.87	(0.49–1.55)	0.643	1.21	(0.54–2.72)	0.651
G8	reduction (vs. maintenance)	0.19	(0.11–0.31)	<0.001	0.12	(0.06–0.25)	<0.001

RFS: recurrence-free survival; OS; overall survival; HR; hazard ratio; CI; confidence interval; MMSE: Mini Mental State Examination; IADL: Instrumental Activity of Daily Living; VES: Vulnerable Elders Survey.

**Table 3 cancers-13-00842-t003:** Perioperative characteristics of HCC patients aged ≥ 70 years stratified according to G8 score change.

Variables	Reduction (*n* = 30)	Maintenance (*n* = 70)	*P*
Age, years	77 (73.0, 84.2)	77 (71.0, 87.6)	0.677
Sex			
male/female	24 (80%)/6 (20%)	51 (73%)/19 (27%)	0.614
Diabetes			
absent/present	22 (73%)/8 (27%)	48 (69%)/22 (31%)	0.812
Esophageal and/or gastric varices			
absent/present	28 (93%)/2 (7%)	64 (91%)/6 (9%)	1.000
HBsAg			
negative/positive	26 (87%)/4 (13%)	60 (86%)/10 (14%)	1.000
HCVAb			
negative/positive	16 (53%)/14 (47%)	44 (63%)/26 (37%)	0.504
Child–Pugh class			
A/B	29 (97%)/1 (3%)	67 (96%)/3 (4%)	1.000
ICGR15, %	16.4 (8.7, 49.1)	18.1 (7.0, 36.1)	0.896
WBC count, 102/μL	46 (32.2, 76.6)	47 (23.4, 80.1)	0.523
Hemoglobin level, g/dL	13.2 (10.8, 15.3)	13.05 (10.6, 15.7)	0.863
Platelet count, ×104/mm3 Pre	17.3 (8.1, 25.9)	15.1 (7.5, 26.1)	0.177
POM6	13.8 (5.6, 28.0)	14.3 (7.3, 20.9)	0.842
Serum albumin level, g/dL Pre	3.8 (3.0, 4.6)	4.0 (3.4, 4.6)	0.183
POM6	3.7 (2.3, 4.6)	4.0 (3.2, 4.5)	0.055
Prothrombin activity, % Pre	85.1 (60.8, 105.2)	86.5 (61.1, 108.3)	0.451
POM6	81.2 (55.5, 111.3)	84.2 (63.7, 103.9)	0.489
Serum total bilirubin level, mg/dL Pre	0.7 (0.4, 1.6)	0.8 (0.4, 1.3)	0.288
POM6	0.8 (0.5, 2.2)	0.8 (0.5, 1.4)	0.612
Creatinine level, mg/dL	0.79 (0.6, 1.1)	0.775 (0.6, 1.1)	0.857
Alpha-fetoprotein level, ng/mL Pre	25.5 (2.2, 19303.2)	9.1 (2.0, 369.7)	0.044
POM6	5.6 (2.3, 152.4)	3.8 (2.0, 23.7)	0.017
PIVKA-II level, mAU/mL Pre	147.5 (14.4, 48583.8)	64 (13.0, 17661.0)	0.299
POM6	19.0 (8.0, 706.1)	18.0 (11.0, 78.8)	0.719
Geriatric Depression Scale score			
<3/≥3	12 (43%)/16 (57%)	34 (50%)/34 (50%)	0.680
Charlson Comorbidity Index			
<4/≥4	15 (54%)/13 (46%)	32 (46%)/38 (54%)	0.632
Vitality index			
<10/≥10	3 (11%)/24 (89%)	15 (23%)/51 (77%)	0.318
G8 score			
<13/≥13	6 (20%)/24 (80%)	28 (40%)/42 (60%)	0.088
Tumor number			
1/2/≥3	25 (83%)/3 (10%)/2 (7%)	55 (79%)/7 (10%)/8 (11%)	0.921
Tumor size, cm	4.3 (0.7, 12.0)	3 (1.1, 8.1)	0.026
Degree of differentiation			
well/moderate/poor or necrotic	7 (23%)/22 (73%)/1 (3%)	13 (19%)/53 (76%)/4 (6%)	0.922
Vascular invasion			
absent/present	9 (30%)/21 (70%)	24 (34%)/46 (66%)	0.853
Surgical margin invasion			
absent/present	24 (89%)/3 (11%)	66 (96%)/3 (4%)	0.345
Nontumor tissue			
normal/chronic hepatitis or liver fibrosis / liver cirrhosis	2 (7%)/24 (80%)/4 (13%)	10 (14%)/45 (64%)/15 (21%)	0.364
Tumor stage			
I/II/III/IVa	2 (7%)/10 (33%)/13 (43%)/5 (17%)	3 (4%)/33 (47%)/30 (43%)/4 (6%)	0.241
Operative procedure			
anatomical resection/non-anatomical resection	16 (53%)/14 (47%)	39 (56%) / 31 (44%)	1.000
Extent of hepatic resection			
≥2 sections/<2 sections	10 (33%)/20 (67%)	18 (26%)/52 (74%)	0.593
Operating time, min	332 (179, 552)	307 (159, 492)	0.442
Operative blood loss, mL	815 (103, 1776)	634 (58, 1543)	0.119
Blood transfusion			
absent/present	24 (80%)/6 (20%)	64 (91%)/6 (9%)	0.175
Complications			
absent/present	24 (80%)/6 (20%)	58 (83%)/12 (17%)	0.955

Data are shown as n (%) or median (5th percentile, 95th percentile). HBsAg: hepatitis B surface antigen; HCV Ab: hepatitis C virus antibody; ICGR15: indocyanine green retention rate at 15 minutes; WBC: white blood cell; POM: post-operative month; PIVKA-II; protein induced by vitamin K absence-II.

**Table 4 cancers-13-00842-t004:** Score change in each of the eight G8 items between groups.

Item	Score	Score Change between Preoperative and 6 Months Postoperative	Reduction (*n* = 30)	Maintenance (*n* = 70)	*P*
1. Has food intake declined over the past 3 months because of loss of appetite, digestive problems, chewing, or swallowing difficulties?	0: severe reduction in food intake, 1: moderate reduction in food intake, 2: normal food intake.	Decline	7	2	0.003
		Stable/increase	23	68	
2. Weight loss during the past 3 months	0: weight loss >3 kg, 1: does not know, 2: weight loss of 1–3 kg, 3: no weight loss.	Decline	14	3	<0.001
		Stable/increase	16	67	
3. Mobility	0: bed or chair bound, 1: can get out of bed/chair but does not go out, 2: goes out	Decline	5	0	0.002
		Stable/increase	25	70	
4. Neuropsychological problems	0: severe dementia or depression, 1: mild dementia or depression, 2: no psychological problems	Decline	2	1	0.213
		Stable/increase	28	69	
5. BMI (weight in kg/height in m2)	0: BMI < 19, 1: BMI 19–21, 2: BMI 21–23, 3: BMI ≥ 23	Decline	8	2	<0.001
		Stable/increase	22	68	
6. Takes >4 medications per day	0: yes, 1: no	Decline	2	4	1.000
		Stable/increase	28	66	
7. Compared with other people of the same age, how does the patient consider his/her health status?	0.0: not as good, 0.5: does not know, 1.0: as good, 2.0: better	Decline	16	2	<0.001
		Stable/increase	14	68	
8. Age	0: >85 years, 1: 80–85 years, 2: <80 years	-	-	-	-
		-	-	-	

**Table 5 cancers-13-00842-t005:** Multivariate Cox regression analysis of recurrence-free survival and overall survival in patients aged ≥70 years with hepatocellular carcinoma who underwent hepatic resection.

Variables	RFS			OS		
	HR	(95% CI)	P	HR	(95% CI)	*P*
Sex female (vs. male)	0.68	(0.29–1.58)	0.369	0.70	(0.24–2.06)	0.513
Age ≥77 years (vs. <77 years)	0.89	(0.42–1.88)	0.761	1.77	(0.49–6.45)	0.384
Serum total bilirubin level ≥0.8 mg/dL (vs. <0.8 mg/dL)	1.26	(0.63–2.52)	0.518	1.05	(0.36–3.08)	0.925
Serum albumin level ≥4.0 g/dL (vs. <4.0 g/dL)	0.73	(0.36–1.47)	0.374	0.29	(0.10–0.85)	0.025
Prothrombin activity ≥86% (vs. <86%)	0.59	(0.30–1.16)	0.128	1.07	(0.43–2.65)	0.882
Platelet count ≥16 × 104/mm3 (vs. <16 × 104/mm^3^)	1.31	(0.65–2.65)	0.449	0.86	(0.31–2.34)	0.763
Alfa-fetoprotein level ≥ 11 ng/mL (vs. <11 ng/mL)	0.99	(0.50–1.97)	0.977	1.27	(0.45–3.58)	0.651
PIVKA-II level ≥73 mAU/mL (vs. <73 mAU/mL)	1.60	(0.80–3.20)	0.186	1.16	(0.35–3.89)	0.807
Tumor number ≥2 (vs. 1)	1.66	(0.79–3.47)	0.178	1.22	(0.36–4.08)	0.752
Tumor size ≥3.5 cm (vs. <3.5 cm)	2.98	(1.39–6.41)	0.005	3.59	(1.12–11.58)	0.032
Surgical margin invasion positive (vs. negative)	4.59	(1.27–16.68)	0.020	0.40	(0.035.63)	0.494
Vitality index maintenance (vs. reduction)	0.88	(0.31-2.50)	0.813	0.50	(0.092.65)	0.413
G8 score maintenance (vs. reduction)	0.16	(0.08–0.34)	<0.001	0.08	(0.03–0.22)	<0.001

RFS: recurrence-free survival; OS; overall survival; HR; hazard ratio; CI; confidence interval; PIVKA-II; protein induced by vitamin K absence-II.

## Data Availability

Data Availability Statement: Data available on request due to restrictions eg privacy or ethical. The data presented in this study are available on request from the corresponding author.

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
