# Peer review of "Perioperative Geriatric Assessment as A Predictor of Long-Term Hepatectomy Outcomes in Elderly Patients with Hepatocellular Carcinoma"

_cancers, 2021, doi:10.3390/cancers13040842_

Round 1

Reviewer 1 Report

cancers-1107150

In this manuscript, the authors evaluated the perioperative geriatric assessment in elderly HCC patients who underwent hepatectomy. They suggested that this assessment, and in particular changes in the G8 score, may be used as a predictor factor for the postoperative outcomes. This is a resubmitted manuscript, as also indicated by the changes made by the authors. I found the manuscript well written and of relevance. In my opinion, no further changes are necessary. The manuscript is ready to be published.

Author Response

Thank you for your valuable comments.

Reviewer 2 Report

I have NO additional comments.

Author Response

Thank you for your valuable comments.

Reviewer 3 Report

The manuscript is written well, but there are still minor problems.

  1. Alpha-fetoprotein levels at POM6 was significantly higher in the G8 reduction group than in the G8 maintenance group. One wonders that the G8 reduction at POM6 may not be a predictive factor for survival but merely be caused as a result of the early recurrence of HCC. The group comparison analysis for RFS and OS excluding patients who developed early recurrence by 6 months would be more persuasive.

  1. The attached table in the cover letter is the same as Table 5 and does not include the baseline G8 score before operation. Please represent the table including it.

Author Response

Thank you for your valuable comments.

Comment 1: Alpha-fetoprotein levels at POM6 was significantly higher in the G8 reduction group than in the G8 maintenance group. One wonders that the G8 reduction at POM6 may not be a predictive factor for survival but merely be caused as a result of the early recurrence of HCC. The group comparison analysis for RFS and OS excluding patients who developed early recurrence by 6 months would be more persuasive.

Response

Recurrence of HCC occurred in 8 patients in the reduction group and in 2 patients in the maintenance group by 6 months after surgery (POM6). Each patient underwent TACE, radiofrequency ablation, or repeat hepatic resection for recurrence of HCC (4 of 10 patients received treatment at 6 months after surgery, and the remaining 6 patients were treated after 6 months). As you have requested, the comparison of survival between the two groups of G8 (G8 reduction and G8 maintenance groups) in 90 patients, excluding the 8 patients in whom HCC recurred, is shown below. This analysis with the excluded patients showed that the maintenance group had a significantly higher RFS and OS than the reduction group.

We have further revised the following sentence in the RESULTS section to explain the patients with recurrent HCC.

RESULTS (line 221-223):

Fig. 2 shows a comparison of long-term outcomes in the G8 score between the two groups. Recurrence of HCC occurred in 8 patients in the reduction group and in 2 patients in the maintenance group by 6 months post-surgery. Each patient underwent transarterial chemoembolization, radiofrequency ablation, or repeat hepatic resection for recurrence of HCC (4 of 10 patients received treatment at 6 months after surgery, and the remaining 6 patients were treated after 6 months). The median follow-up periods were 50.5 and 24.1 months in the maintenance and reduction groups, respectively. The 5-year RFS rates were 43.4% and 6.7% in the maintenance and reduction groups, respectively (hazard ratio, 5.35; 95% CI, 3.18–9.01; P<0.001; Fig 2A). The 5-year OS rates were 73.8% and 17.8% in the maintenance and reduction groups, respectively (hazard ratio, 8.09; 95% CI: 4.03–16.27; P<0.001; Fig 2B).

Comment 2: The attached table in the cover letter is the same as Table 5 and does not include the baseline G8 score before operation. Please represent the table including it.

Response

Apologies for any confusion caused. The result of the examination with G8 score before the operation examined is shown below and we have updated this in the manuscript (Table 5; lines 277-279).

Based on your previous comment, we evaluated the preoperative G8 score by multivariate Cox regression analysis of recurrence-free survival and overall survival. Please see the table below. We first focused on the preoperative geriatric assessment score as you pointed out. We found that the preoperative G8 score turned out to be a prognostic factor for overall survival, but not for recurrence-free survival. Therefore, we focused on the rate of change of each geriatric assessment score from before to 6 months after surgery.

This manuscript is a resubmission of an earlier submission. The following is a list of the peer review reports and author responses from that submission.

Round 1

Reviewer 1 Report

cancers-1019640

In this manuscript, the authors analyzed the complete geriatric assessment as a predictor for the recurrence free survival and the overall free survival in elderly HCC patients who underwent hepatectomy. Particularly, the authors highlighted that changes in the G8 score are useful to predict postoperative outcomes in these patients.

Although the manuscript is well written, some issues are still unresolved.

  1. The study analyze a group of 100 patients, in which only ¼ are female. I wonder if this limitation is due to the low number of elderly female patients with HCC or to a bias in the patients’ recruitment.
  2. Table 2. Although it is well explained in the material and methods section, I suggest to include a brief description of the reduction/maintenance group also in the results to facilitate the comprehension of the analysis performed.
  3. Table 3. Authors state that the patients are stratified according to G8 score change. However, from material and methods, I understood (maybe incorrectly!) that the stratification (reduction vs maintenance) was based on CGA components rather than on G8 score.
  4. Table 3. G8 score. In the material and methods section, authors refer to <14 as a geriatric risk profile. However, in this table, they choose a cut off of 13. The same occurs for the GDS-scale (3 instead of 5). Please, can you explain this?
  5. It is not clear if even other components of the GSA vary between the preoperative and the post-operative state. Please, add it to the text.
  6. Table 4. Please, indicate the type of statistical test used.
  7. Although the results support the discussion, I found this section too focus on G score. Other important results founf are not mentioned.
  8. Line 189. Please change “nutrition” with “nutritional status”.

Author Response

Responses to the comments of Reviewer #1

Thank you for your valuable comments.

  1. The study analyze a group of 100 patients, in which only ¼ are female. I wonder if this limitation is due to the low number of elderly female patients with HCC or to a bias in the patients’ recruitment.

Response

Our study showed the male:female ratio for HCC was 3.17:1, and this result occurred by chance and was not planned. Our ratio was similar to the ratio of 2.26:1 reported by Kudo et al. from a large study. In that study, the 20th Nationwide Follow-up Survey of Primary Liver Cancer in Japan, they found that compared to the 19th survey, patients with HCC were older at the time of clinical diagnosis, included more female patients, included more patients with non-B, non-C HCC, had smaller tumor diameters, and received radiofrequency ablation more frequently than local ablation therapy. This survey included data from 21,075 new patients and 40,769 previously followed patients, which were compiled from 544 institutions over a 2-year period from 1 January 2008 to 31 December 2009 [Kudo M, Izumi N, Kubo S, et al.: Report of the 20th Nationwide follow-up survey of primary liver cancer in Japan. Hepatol Res. 2020; 50: 15-46.]. Therefore, this ratio appears to be common across multiple populations studied.

  1. Table 2. Although it is well explained in the material and methods section, I suggest to include a brief description of the reduction/maintenance group also in the results to facilitate the comprehension of the analysis performed.

Response

According to your comments, we have inserted the following sentences in the RESULTS section.

RESULTS (page 7, line 120-125):

Changes in perioperative geriatric assessments

Table 2 shows the results of a multivariate statistical model incorporating perioperative changes in eight geriatric assessments. The rate of change in each score between preoperative and postoperative assessments was calculated by subtracting the preoperative score from the score at 6 months postoperatively, then dividing by the score at 6 months postoperatively. Patients with score change ≥0 comprised the maintenance group, while patients with score change <0 comprised the reduction group.

  1. Table 3. Authors state that the patients are stratified according to G8 score change. However, from material and methods, I understood (maybe incorrectly!) that the stratification (reduction vs maintenance) was based on CGA components rather than on G8 score.

Response

Multivariate Cox proportional hazards analysis identified the G8 change as an independent prognostic predictor for RFS and OS (Table 2). Therefore, we have classified our patients in two groups according to the G8 change, but not to the changes of other CGA components. We have inserted the following sentence in the MATERIALS AND METHODS section.

MATERIALS AND METHODS (page 24, line 384-385):

We classified the perioperative characteristics, including the CGA components of HCC patients of the two groups, according to the G8 change.

  1. Table 3. G8 score. In the material and methods section, authors refer to <14 as a geriatric risk profile. However, in this table, they choose a cut off of 13. The same occurs for the GDS-scale (3 instead of 5). Please, can you explain this?

Response

Each of these preoperative items (GDS, comorbidity index, vitality index, G8) was classified into two groups using the median value from our 100 patients. In response to your comments, we have now clarified this by inserting the following sentence in the MATERIALS AND METHODS section.

MATERIALS AND METHODS (page 23, line 376-378):

All of these preoperative items were classified into two groups using the median value of our 100 patients.

  1. It is not clear if even other components of the GSA vary between the preoperative and the post-operative state. Please, add it to the text.

Response

Based on your comments, we evaluated the perioperative changes in an additional seven geriatric assessments. We have now created a new Fig. 1 and inserted the following sentence in the RESULTS section.

RESULTS (page 7, line 112-119):

Changes in perioperative geriatric assessments

Fig. 1 shows perioperative changes in eight geriatric assessments between the maintenance and reduction groups. There was a significant difference between the two groups in all assessments, except for the MMSE. The largest difference among all assessments was found in the G8 score between the two groups. In the reduction group, the mean G8 score decreased from 13.8 preoperatively to 11.4 at 6 months postoperatively, compared to the maintenance group, in which the mean G8 score increased from 13.0 to 14.5 during that same time period; the difference between groups at the 6 month-postoperative time point was statistically significant (p<0.001; Fig. 1H).

  1. Table 4. Please, indicate the type of statistical test used.

Response

Please note that the description of the statistical test used is included in the Statistical analysis section of the MATERIALS AND METHODS.

MATERIALS AND METHODS (page 25, line 414-415):

Differences between groups concerning perioperative changes in eight geriatric assessments were assessed using the chi-squared test or Fisher’s exact test.

  1. Although the results support the discussion, I found this section too focus on G score. Other important results founf are not mentioned.

Response

In response to your comments, we have now added the following sentences to the Discussion section.

DISCUSSION (page 19, line 236-245):

The group with the reduced postoperative G8 score had a greater median serum AFP concentration and a larger tumor size (Table 3). Many researchers, including us, have identified preoperative serum AFP concentration and tumor size as prognostic factors for recurrence and survival after surgery for HCC [18, 19]. Therefore, we consider it necessary to re-evaluate the results of this study using statistical methods such as propensity score matching or inverse probability of treatment weighting in order to eliminate bias between the two groups. However, this type of examination was not possible due to the small number of patients in each group in our study. Therefore, in the future, we plan to examine a larger cohort as a multicenter collaborative-research study.

  1. Line 189. Please change “nutrition” with “nutritional status”.

Response

We have now revised “nutrition” to “nutritional status” on line 189 (new line 250).

Reviewer 2 Report

This retrospective study of Kaibori et al demonstrated that the change in Geriatric 8 (G8) score at 6 months post-operation, which was partially based on nutritional assessment, was the most significant predictive factor for RFS and OS among the tested geriatric assessments, in 100 consecutive HCC patients aged ≥70 years who qualified for elective liver resection. The authors concluded that perioperative changes in G8 score could help forecast postoperative long term outcomes in these patients.

Major points

  1. Figure 1A shows that HCC recurred in some patients before 6 months post-operation, which was the point of Comprehensive Geriatric Assessment (CGA). If retreatment was given for such patients before the time-point, the intervention could affect the results of CGA. Such patients retreated within 6 months should be excluded.
  2. The results may have been biased by tumor factors, especially size of tumor, which was significant in Tables 3 & 5. I wonder whether the maintained G8 score was still a significant factor for RFS and OS, even if patients were divide into two groups according to tumor size of ≥ 3.5 cm or < 3.5 cm.
  3. The OS should be affected by whether curative retreatment could be carried out for recurred HCC. The choice of retreatment should have been based, at least in part, on patients’ general condition, including nutritional status. The authors should discuss this point.

Minor points

  1. In Table 1, the author should mention how they determined the cutoff value of each preoperative geriatric assessment. For example, the proportion of patients with G8 score <13 was 34%, while the proportion of patients with Barthel index = 100 was 82%; the proportions varies so widely. With the other cutoff values, the results may be different.
  2. In Table 1, 30% of patients had a history of other cancers. The author should show a few examples of cancer type in text.
  3. The results of a univariate should be shown in Tables 2 & 5.
  4. Alfa-fetoprotein is abbreviated sometime, and spelled out the other time. Please check.
  5. In the Discussion section, reference #10 is old [WJG 2006]. Please cite the newer guidelines.
  6. In the Methods section, the 100 patients have been selected from 230 patients with underwent R0 resections. The author should show a flowchart as a supplementary material.
  7. Patients underwent CGA measurements … conducted by trained research assistants (lines 242-243). How many assistants were involved? How can the authors know if consistent data was obtained by different assistants?
  8. The treatment strategies for HCC were shown in the Methods section (lines 288-295). The author should show on which guideline they were based, because the guidelines have recently been updated almost year by year with the rapid progress in this field.

Author Response

Responses to the comments of Reviewer #2

Thank you for your valuable comments.

Major points

  1. Figure 1A shows that HCC recurred in some patients before 6 months post-operation, which was the point of Comprehensive Geriatric Assessment (CGA). If retreatment was given for such patients before the time-point, the intervention could affect the results of CGA. Such patients retreated within 6 months should be excluded.

Response

Recurrence of HCC occurred in 8 patients in the reduction group and in 2 patients in the maintenance group by 6 months after surgery. Each patient underwent TACE, radiofrequency ablation, or repeat hepatic resection for recurrence of HCC (4 of 10 patients received treatment at 6 months after surgery). As you have pointed out, the comparison of survival between the two groups of G8 in 90 patients, excluding the 8 patients in whom HCC recurred, is shown below. The maintenance group had a significantly higher RFS and OS than the reduction group.

At first, we focused on the preoperative geriatric assessment score, as per your comment. However, we found that the preoperative G8 score was a prognostic factor for overall survival, but not for recurrence-free survival. Therefore, we focused on the rate of change of each geriatric assessment score from before to 6 months after surgery.

The primary aim of our study was to examine the correlation between the change from preoperative to postoperative CGA and prognosis. Based on the above, we would like to present the current results of this study, including data from patients with recurrent HCC.

We have inserted the following sentence in the RESULTS section.

RESULTS (page 9, line 140-150):

Fig. 2 shows a comparison of long-term outcomes in the G8 score between the two groups. Recurrence of HCC occured in 8 patients in the reduction group and in 2 patients in the maintenance group by 6 months post-surgery. Each patient underwent transarterial chemoembolization, radiofrequency ablation, or repeat hepatic resection for recurrence of HCC (4 of 10 patients received treatment at 6 months after surgery, and the remaining six patients were treated after 6 months). The median follow-up periods were 50.5 and 24.1 months in the maintenance and reduction groups, respectively. The 5-year RFS rates were 43.4% and 6.7% in the maintenance and reduction groups, respectively (hazard ratio, 5.35; 95% CI, 3.18–9.01; P<0.001; Fig 2A). The 5-year OS rates were 73.8% and 17.8% in the maintenance and reduction groups, respectively (hazard ratio, 8.09; 95% CI: 4.03–16.27; P<0.001; Fig 2B).

  1. The results may have been biased by tumor factors, especially size of tumor, which was significant in Tables 3 & 5. I wonder whether the maintained G8 score was still a significant factor for RFS and OS, even if patients were divide into two groups according to tumor size of ≥ 3.5 cm or < 3.5 cm.

Response

Tumor size is considered to be one of the confounding factors for RFS and OS. Therefore, we used multivariate Cox regression analysis to clarify the causal relationship of G8 by eliminating the effects of confounding factors such as tumor size (Table 5).

  1. The OS should be affected by whether curative retreatment could be carried out for recurred HCC. The choice of retreatment should have been based, at least in part, on patients’ general condition, including nutritional status. The authors should discuss this point.

Response

As per your suggestion, we have now revised and inserted the following sentences in the DISCUSSION section.

DISCUSSION (page 20, line 267-275):

In this study, HCC recurred in 8 patients in the reduction group and 2 patients in the maintenance group by 6 months after surgery. Each patient underwent transarterial chemoembolization, radiofrequency ablation, or repeat hepatic resection for recurrence of HCC (4 of 10 patients received treatment at 6 months after surgery). The OS is affected by the ability to perform curative treatment for recurrent HCC. Treatments for recurrent HCC may worsen the patient's general condition, including their nutritional status. It is therefore important to note that our data include deterioration of postoperative G8 values due to recurrence of HCC and its treatment. Further studies of postoperative survival in patients with HCC who undergo liver resection are needed to confirm our findings.

Minor points

  1. In Table 1, the author should mention how they determined the cutoff value of each preoperative geriatric assessment. For example, the proportion of patients with G8 score <13 was 34%, while the proportion of patients with Barthel index = 100 was 82%; the proportions varies so widely. With the other cutoff values, the results may be different.

Response

Each of the preoperative items (GDS, comorbidity index, vitality index, G8) were classified into two groups using the median value from our 100 patients. Based on your comment, we have now inserted the following sentence in the MATERIALS AND METHODS section.

MATERIALS AND METHODS (page 23, line 376-378):

All of these preoperative items were classified into two groups using the median value of our 100 patients.

  1. In Table 1, 30% of patients had a history of other cancers. The author should show a few examples of cancer type in text.

Response

According to your comments, we have inserted the following sentences in the RESULTS section.

RESULTS (page 4, line 92-95):

The history of cancer in these 30 patients was as follows: gastric cancer (5 cases), colon cancer (5 cases), prostate cancer (5 cases), bladder cancer (3 cases), lung cancer (2 cases), pancreatic cancer (2 cases), laryngeal cancer (2 cases), and other cancers (6 cases).

  1. The results of a univariate should be shown in Tables 2 & 5.

Response

Table 2 is a univariate analysis. Table 5 is a multivariate analysis of items that are important in relation to survival outcomes (We don't think univariate regression analysis is needed). This paper is a study that explores causal relationships, and we think that how to eliminate confounding factors affects the quality of the study. Therefore, we believe that analysis by multiple regression analysis including confounding that can be considered from clinical factors is necessary, and that univariate regression analysis is unnecessary.

  1. Alfa-fetoprotein is abbreviated sometime, and spelled out the other time. Please check.

Response

We have now listed AFP as an abbreviation for alpha-fetoprotein in the text.

  1. In the Discussion section, reference #10 is old [WJG 2006]. Please cite the newer guidelines.

Response

Based on your suggestion, we have now revised to reference the newer guidelines.

New ref. No. 10.

Kokudo N, Takemura N, Hasegawa K, et al. Clinical practice guidelines for hepatocellular carcinoma: The Japan Society of Hepatology 2017 (4th JSH-HCC guidelines) 2019 update. Hepatol Res. 2019; 49: 1109-1113.

  1. In the Methods section, the 100 patients have been selected from 230 patients with underwent R0 resections. The author should show a flowchart as a supplementary material.

Response

Based on your suggestion, we have created a flowchart of how the patients were selected, and this will be included as a supplementary figure (Supplemental Digital Content 1).

  1. Patients underwent CGA measurements … conducted by trained research assistants (lines 242-243). How many assistants were involved? How can the authors know if consistent data was obtained by different assistants?

Response

We have deleted this sentence from the MATERIALS AND METHODS section. All surgeons and the hospital care team members were blinded to CGA results. We have now altered the following sentence in the MATERIALS AND METHODS section.

MATERIALS AND METHODS (page 22, line 337-340):

Patients underwent CGA measurements both preoperatively and at 6 months postoperatively. These CGA studies were conducted by two medical assistants who were familiar with medical information. Any contents of the GA that could not be confirmed at the time of data collection by the assistant were validated by the attending physician, who determined the final score.

  1. The treatment strategies for HCC were shown in the Methods section (lines 288-295). The author should show on which guideline they were based, because the guidelines have recently been updated almost year by year with the rapid progress in this field.

Response

According to your comments, we have inserted the following sentences in the MATERIALS AND METHODS section.

MATERIALS AND METHODS (page 21, line 303-320):

Clinicopathologic variables, treatment algorithm for HCC, and surgical procedures

Before surgery, each patient underwent conventional liver function tests and measurement of the indocyanine green retention rate at 15 min (ICGR15). Hepatitis screening was performed by measuring hepatitis B surface antigen and hepatitis C antibody. Levels of AFP and protein induced by vitamin K absence/antagonism-II (PIVKA-II) were also measured in all patients.

We used the updated treatment algorithm for HCC based on a combination of the following five factors: liver function reserve, extrahepatic metastasis, vascular invasion, tumor number, and tumor size [10]. Liver functional reserve was evaluated based on the Child-Pugh classification, and when hepatectomy was being considered, a decision was made based on the degree of liver damage, including a measurement of ICGR15. The new treatment algorithm is summarized as follows. One of three treatment regimens is recommended for HCC patients with Child-Pugh A/B liver function without extrahepatic metastasis or vascular invasion. First, either surgical resection radiofrequency ablation is recommended with no priority for up to three HCCs measuring ≤3cm, or surgical resection is recommended as first-line therapy for solitary HCC regardless of size. Second, for up to three HCCs measuring >3cm, surgical resection is recommend as first-line therapy, and transarterial chemoembolization is recommended as second-line therapy. Third, for patients with HCC accompanied by vascular invasion without extrahepatic metastasis, a combination of embolization, hepatectomy, hepatic arterial infusion chemotherapy, and molecular targeted therapy are recommended. Treatment is selected for each patient according to the individual situation, including consideration of the following factors: liver function, the condition of HCC, and the extent of vascular invasion.

Author Response

Responses to the comments of Reviewer #3

Thank you for your valuable comments.

  1. Table 3 shows the baseline and perioperative patient characteristics according to the chances in the G8 score before and after operation. In this study, however, the G8 score was evaluated before and 6 months after operation in each patient. Table 3 should include the parameter about the patient condition up to 6 months after operation. In particular, the tumor status (including tumor markers) and hepatic functional reserve up to 6 months after operation should be compared between "reduction" and "maintenance" groups.

Response

Based on your comments, we evaluated platelet count, prothrombin activity, serum albumin, total bilirubin, AFP, and PIVKA-2 levels and compared them between the two groups before surgery and 6 months after surgery. We have added and modified these results to Table 3.

RESULTS (page 9, line 152-161)

Perioperative characteristics of HCC patients classified according to G8 change

Table 3 summarizes the perioperative characteristics of both groups. Platelet count, prothrombin activity, serum albumin, total bilirubin, alpha-fetoprotein (AFP), and PIVKA-2 levels were compared between the two groups before and 6 months after surgery. No differences were detected between groups in terms of age, sex, diabetes mellitus status, presence or absence of esophageal and/or gastric varices, hepatitis B surface antigen status, hepatitis C virus antibody, Child–Pugh class, indocyanine green retention rate at 15 min, peripheral blood count, general biochemical blood laboratory test results, conventional liver function test results, or serum PIVKA-II concentrations. The serum AFP concentration significantly differed between groups both before surgery and 6 months after surgery.

  1. Table 5; it would be better that the baseline G8 score before operation is included as an explanatory variable.

Response

Based on your comment, we evaluated the preoperative G8 score by multivariate Cox regression analysis of recurrence-free survival and overall survival. Please see the table below. We first focused on the preoperative geriatric assessment score as you pointed out. We found that the preoperative G8 score turned out to be a prognostic factor for overall survival, but not for recurrence-free survival. Therefore, we focused on the rate of change of each geriatric assessment score from before to 6 months after surgery.
